# Sex Differences in the Association between Risk of Anterior Cruciate Ligament Rupture and COL5A1 Polymorphisms in Elite Footballers

**DOI:** 10.3390/genes14010033

**Published:** 2022-12-22

**Authors:** Gil Rodas, Alejandro Cáceres, Eva Ferrer, Laura Balagué-Dobón, Lourdes Osaba, Alejandro Lucia, Juan R. González

**Affiliations:** 1Medical Department of Football Club Barcelona (FIFA Medical Centre of Excellence), 08970 Barcelona, Spain; 2Barça Innovation Hub of Football Club Barcelona, 08028 Barcelona, Spain; 3Sports Medicine Unit, Hospital Clínic-Sant Joan de Déu, 08029 Barcelona, Spain; 4ISGlobal, 08003 Barcelona, Spain; 5Centro de Investigación Biomédica en Red en Epidemiología y Salud Pública (CIBERESP), 08003 Barcelona, Spain; 6Department of Mathematics, Escola d’Enginyeria de Barcelona Est (EEBE), Universitat Politècnica de Catalunya, 08019 Barcelona, Spain; 7Progenika Biopharma, A Grifols Company, 48160 Derio, Spain; 8Faculty of Sport Sciences, Universidad Europea de Madrid, 28670 Villaviciosa de Odón, Spain; 9Research Institute Imas12, Hospital 12 de Octubre, 28041 Madrid, Spain; 10Department of Mathematics, Universitat Autònoma de Barcelona, 08193 Barcelona, Spain

**Keywords:** anterior cruciate ligament, injury, single-nucleotide polymorphisms, female, team sport, football, collagen, *COL5A1*, rs13946, rs12722, sex differences

## Abstract

Background: Single-nucleotide polymorphisms (SNPs) in collagen genes are predisposing factors for anterior cruciate ligament (ACL) rupture. Although these events are more frequent in females, the sex-specific risk of reported SNPs has not been evaluated. Purpose: We aimed to assess the sex-specific risk of historic non-contact ACL rupture considering candidate SNPs in genes previously associated with muscle, tendon, ligament and ACL injury in elite footballers. Study Design: This was a cohort genetic association study. Methods: Forty-six (twenty-four females) footballers playing for the first team of FC Barcelona (Spain) during the 2020–21 season were included in the study. We evaluated the association between a history of non-contact ACL rupture before July 2022 and 108 selected SNPs, stratified by sex. SNPs with nominally significant associations in one sex were then tested for their interactions with sex on ACL. Results: Seven female (29%) and one male (4%) participants had experienced non-contact ACL rupture during their professional football career before the last date of observation. We found a significant association between the rs13946 C/C genotype and ACL injury in women footballers (*p* = 0.017). No significant associations were found in male footballers. The interaction between rs13946 and sex was significant (*p* = 0.027). We found that the C-allele of rs13946 was exclusive to one haplotype of five SNPs spanning *COL5A1*. Conclusions: The present study suggests the role of SNPs in genes encoding for collagens as female risk factors for ACL injury in football players. Clinical Relevance: The genetic profiling of athletes at high risk of ACL rupture can contribute to sex-specific strategies for injury prevention in footballers.

## 1. Introduction

Athletes continuously push the boundaries of human movement; therefore, they are often exposed to injury [1]. Unintended sideways motions can snap the ligaments that stabilize the knee, one of the most complex joints in the human body. The anterior cruciate ligament (ACL) resists the excessive rotations of the knee and movements of the tibia to allow efficient bipedal locomotion to be achieved, holding the thigh bone in place on top of the shin bone. The ligament is particularly at risk of rupture in sports, such as football (soccer), with sudden changes in the direction of movement [2]. ACL rupture is immediately disabling, requiring surgery in 90% of athletes [3]. This injury has enduring consequences for the health of patients, including an increase in the likelihood of osteoarthritis (71%) [4], in addition to the immediate loss of competition opportunities for the professional athlete and the inability to regain their preinjury level in 50% of cases [5]. Strategies for ACL rupture prevention are, therefore, crucial for reducing the impact of severe injury and helping to protect the health and work of sport practitioners and the general community. 

Important risk factors for non-contact ACL rupture include sport, sex and genetics. Football (soccer) remains one of the highest-risk sports for this type of injury [2], with an estimated relative risk of 1.2% over a period from 1 season to 4 years [6]. Football is practiced more by men than women; however, female participation has seen incredible growth in recent years and is expected to reach 60 million practitioners in 2026 [7]. Given the popularity of the sport and the expected growth of female participation, the incidence of ACL rupture is expected to increase substantially in the near future. With the current male-biased participation in football, the incidence appears to be similar between the sexes [8]. However, as the incidence rate ratio is 2.2-fold higher in female footballers than in males [8,9,10], the increase in ACL injuries will be particularly pronounced as more women take up the sport. Professional female football has also witnessed great gains in public exposure and revenue, making elite female footballers a particular population at risk of ACL rupture. Surgery is more likely to be recommended in female athletes with ACL tears, as the female knee may be more cruciate dependent than the male knee [11]. There is a need for clinical studies to segregate sexes to assess female-specific risks and interventions in tendon and ligament injuries [12,13,14]. As such, particular efforts are needed to design specific strategies for ACL rupture prevention in female footballers to safely promote their expected growth in number and playing hours. 

The constituents of the human knee form a complex locomotive mechanism that gave bipedal hominids a crucial evolutionary advantage; therefore, they have been subjected to strong selection pressures [15]. In this context, ACL function and rupture are highly sensitive to genetic variation. The estimated heritability of the injury in twin studies is 69% [16], and genetic studies have identified associations with single-nucleotide polymorphisms (SNPs). A recent study of ACL injury in the general population observed three SNPs at a genome-wide significant level and replicated a previous association in a collagen gene, *COL3A1* [17]. 

Much research has focused on the susceptibility variants to injuries in collagen genes *COL5A1* and *COL1A1*, albeit with mixed results [18]. The interest in collagen is based on the observations that as a structural component of ligament fibrils, it is associated with joint laxity, a risk factor for non-contact ACL injury in both men and women [19]. When stratified by sex, *COL5A1* variant rs12722 was associated with genu recurvatum and general joint laxity in females [20]. As joint laxity is typically higher in women and likely modulated by sex hormones [21], these observations offer a window on the differences in the risk of ACL rupture between sexes. Associations between injury and variants in collagen genes in female athletes at high risk have not been fully confirmed, either due to lack of power or biases [22]. In general population studies, associations still need to be disaggregated by sex and selected by high-risk activity. In addition, association studies on very few SNPs may miss the causal variant. Accordingly, genetic associations studies that test the differences in the risk of ACL rupture between sexes are needed in a homogenous cohort of high-risk athletes. 

The aim of the present prospective cohort study was to assess the association between previously identified candidate SNPs in genes encoding for collagen and the risk of ACL injury in a population of elite professional female and male players from a high-risk team sport (i.e., soccer). The population was homogeneous with regards to age, fitness level and athletic performance.

## 2. Materials and Methods

### 2.1. Participants

The study was conducted with players of the first division of the Spanish National League in both men’s (*LaLiga Santander*) and women’s (*Primera Iberdrola*) categories who played for FC Barcelona during the 2020–2021 season. The population sample included the best male and female football players in the world, according to *Fédération Internationale de Football Association* (FIFA). Both male and female teams were top-performance teams whose players have won the most important competitions in the sport. 

We recorded the history of non-contact ACL rupture of all players from FC Barcelona using a validated electronic medical record system (Gem version 1.2; FCB, Spain). Injury diagnoses were made by the relevant medical physician (team doctor) during the evaluation period, who also retrieved detailed information about previous ACL injuries of the participants occurred before joining the club. Regarding ancestry, 23 female football players were of European origin and one of African ancestry, whereas males were more diverse, with 11 being of European ancestry, 3 of African origin and 8 from Latin America of likely admixture origin.

### 2.2. Ethics Statement

The study was conducted according to the guidelines of the Declaration of Helsinki and was approved by the Ethics Committee of Consell Català del’Esport, (01/2018CEICGC). All participants were informed of the risks and benefits of the study and gave written consent for genotyping. All personal information and results were anonymized.

### 2.3. Genotyping 

The results of a previous GWAS study [23] were complemented by a comprehensive literature review to select 108 SNPs that have been associated with muscle, tendinopathy and ligament injuries. The list of the SNPs, and their genomic position and location within each candidate gene are shown in Appendix A. Blood samples from each participant were collected into EDTA vacutainer tubes. Genomic DNA was isolated using a QIAamp DNA Blood Mini kit (Qiagen, Hilden, Germany) at Synlab laboratory (SYNLAB Diagnósticos Globales S.A.U., Esplugues de Llobregat, Spain), and genotyping was performed using Kompetitive Allele Specific PCR genotyping technology (KASP™, Hoddesdon, UK) at Progenika laboratories (Derio, Spain). 

### 2.4. Statistical Analysis

Genotype calling was performed for all the samples and SNPs, on which we performed quality control tests. For each SNP, we tested for percentage of genotype calling and determined whether the genotype distribution met the Hardy–Weinberg Equilibrium (HWE). Descriptive statistics were calculated for each genotype. Differences among genotypes, allelic frequencies and *p*-values for the HWE are shown in Appendix A. We performed genetic association analyses using dominant, recessive and additive models using the *SNPassoc* R package (version 2.0-11) [24] and performed haplotype inferences with *haplo.stats* (version 1.8.7) [25]. Since the SNPs were selected from previous evidence and the number of individuals of our study was low, we reported nominal statistical significance, as multiple comparison correction is used to discover associations with no previous evidence and can hinder statistical power [26]. We used the exact distribution of max-statistic (e.g., the most significant *p*-value obtained from dominant, recessive and additive tests) to compute the *p*-value of association corrected for the number of inheritance models [27].

We meta-analyzed the risk of ACL for men and women using our data and those reported in the meta-analysis by Guo et al. [28] from eight different studies. As reported, the studies had significant heterogeneity. Therefore, for adding our associations to the meta-analysis, we used Fisher’s method for the combination of *p*-values with the *poolr* R package (version 1.1-1). 

## 3. Results

Forty-six professional (twenty-four women and twenty-two men) football players were recruited for the study from the first team of FC Barcelona, season 2020-21. Seven female players and one male player had experienced non-contact ACL rupture before the time of the last observation (July 2022). We observed that the risk of injury was 8.3 times higher for females than for males (odds ratio = 8.29, Fisher’s exact test *p* = 0.048). 

Of the 108 SNPs, 92 were considered for downstream analyses, as 16 (12.9%) were removed due to problems with the HWE (*p* < 0.01), the missing rate (>5%) and low minor allele frequency (5%). The mean of the major allele frequency of the analyzed SNPs was 72.6%, ranging from 50.0% to 93.8%. We inspected the correlation structure of the SNPs, as many belonged to the same gene. We observed complete linkage disequilibrium in rs5745697 and rs5745678 in hepatocyte growth factor gene (*HGF*) and among rs13946, rs16399, rs1134170, rs55748801 and rs71746744 in *COL5A1*.

We tested for associations between the SNPs and ACL rupture in females using three genetic models (recessive, dominant and additive). We found a nominally significant association with the recessive model of the C-allele of rs13946 (*p* = 0.017) and likewise for all the SNPs in *COL5A1* and in complete linkage disequilibrium. The association was also significant after adjusting for all the models tested (MAX-statistic = 5.61, *p* = 0.035). More specifically, for female footballers with no injuries, we observed 13 with genotype T/T, 4 with C/T and 0 with C/C. By contrast, ACL injuries appeared in four female footballers with T/T and in three with C/C, thereby suggesting that the homozygous group of the C-allele was a risk factor (Table 1). We also observed a significant association with the dominant model of the A-allele of rs3196378 within *COL5A1* (*p* = 0.040). The distribution of alleles in the injured groups was similar to that for rs13946; however, the minor allele frequency was higher in the non-injured group, and the association was not significant after correction for the type of model. 

We performed a similar analysis on rs12722 in *COL5A1*, which has also been extensively investigated with respect to ACL injury [28]. We observed six, nine and two female footballers with T/T, C/T and C/C alleles and no injuries, respectively, whereas we found three, one and three injuries for T/T, C/T and C/C. The association with the recessive model was the strongest; it was, however, not significant (*p* = 0.107). The same analysis on males revealed no significant associations across the 108 SNPs, most likely due to the low number of ACL injuries. 

For the significant association with rs13946 in females, we performed an interaction analysis between the additive model and sex for ACL injury. We observed a significant change between sexes in the association between the C-allele and risk of ACL (*p* = 0.027) (Figure 1A), suggesting a female-specific risk with the presence of the C-allele. 

As the association between ACL injury and rs13946 has been previously reported [28], we assessed the consistency of our results with those discussed in a recent meta-analysis by Guo et al. [28]. In the reported meta-analysis, the frequencies of the alleles conditioned by ACL injury were available for all studies, together with information about whether the studies were on males, females or both. From the seven studies that assessed associations with rs13946, only one was on female athletes [22], where no significant association was reported. We reassessed the recessive model of the C-allele in that study and found no significant associations (*p* = 0.96). We then combined these results with ours, using Fisher’s method for the combination of *p*-values, and observed a trend of significance (*p* = 0.08). We also considered the recessive model combining the data of both males and females in our study. We observed a significant association between the C/C homozygous genotype for rs13946 and ACL injury in all 46 footballers. We meta-analyzed this result with those from both sexes as described by Guo et al. [28]. We reassessed the recessive model on five studies considered in the reported meta-analysis [28] and combined the results with ours for both males and females. We failed to observe a significant association after *p*-value combination (*p* = 0.86). The results suggest that female-specific associations should be considered when assessing the risk of SNPs in *COL5A1*.

As we observed strong linkage disequilibrium across the SNPs in *COL5A1*, we performed a haplotype analysis. We inferred the most likely haplotypes across SNPs rs12722, rs13946, rs3196378, rs16399 and rs1134170 for each individual in our study. A list of possible haplotypes was inferred with the EM algorithm implemented in *haplo.stats*, and probabilities for the observation of particular haplotypes were assigned. Possible haplotype pairs for each individual were listed, and the pairs with the highest probabilities in the sample were assigned to the individual. We inferred four different haplotype groups in female footballers (h1, h2, h3 and h4). Figure 1B shows all 48 haplotypes inferred in the 24 female players. Notably, we observed that haplotype h1 was the only one that included the C-allele of rs13946. As shown in Figure 1B, there was an over-representation of injuries for this haplotype (h1 in red) that matched exactly that for the C-allele of rs13946 and thus the association with ACL injury. We also inferred the haplotypes in male footballers (Figure 1B). While males had more haplotype diversity, with six haplotype groups, we observed again that the C-allele only appeared in the h1 group. Therefore, the interaction analysis on the h1 haplotype and sex gave results identical to those for rs13946. These results suggest that the female-specific risk of ACL in *COL5A1* might be supported by an extended haplotype spanning the gene, which should also be considered when assessing the role of the genetic variability of the gene in ACL occurrence.

## 4. Discussion

We studied the occurrence of non-contact ACL injury in a group of individuals at extreme risk. In contrast to the general population, the increased risk in our study group was related to their professional activity and, in particular, to the continuous demand to push their limits of achievement and performance. Our study was highly controlled. All participants belonged to the first team of the same football club during the same season, with comparable levels of sporting success between male and female teams, and all were subjected to the same demands with similar training and medical facilities. We observed that ACL rupture was 8.3 times greater in female players than in male counterparts. We thus not only confirmed the higher female-associated risk of this injury [29], but we also found that the observed difference in risk between sexes was particularly high, even when compared with footballers in general [8]. This observation is in line with the expected extreme risk of our study sample.

We explored the underlying genetic variability that could help to explain the large difference in ACL risk between sexes with association tests using a comprehensive set of polymorphic variants previously associated with musculoskeletal injuries, but we only observed significant associations with SNPs in *COL5A1*. Several genes included in our study have been previously associated with ACL rupture both in the general population and in football practitioners. We failed to find significant associations with *ACTN3* (rs1815739), *ACAN* (rs1516797) and *VEGFA* (rs2010963), all previously found to be associated with ACL rupture in footballers [18], or with rs1800255 in *COL3A1*, validated in a large population sample [17]. Differences in sample size, ACL evaluation and diversity of the populations in the studies are the likely causes of these disparities. While our study was limited by its small sample size, the individuals were highly homogenous and were at extreme risk. As such, the genetic associations may differ from those found in more general cohorts.

We observed associations between ACL rupture and *COL5A1* (rs13946) that appeared to be specific to female footballers. This gene has been examined in-depth for its role in ACL injures. *COL5A1* encodes for the collagen α-1(V) chain, which has been associated with differences in ligament laxity between sexes [20]. We found that homozygous C/C in rs13946 was a risk factor only for women. Remarkably, in a recent meta-analysis, the C-allele appeared to be protective against injury, after controlling for study heterogeneity [28]. We would like to note that most of the studies assessing the risk of rs13946 failed to stratify by sex, and only one was specific to females [22]. Combining our results with the aforementioned study revealed a trend of significance for the recessive model of rs13946, whereas the combination of our study (including males and females) with five other studies on both sexes failed to reach significance. These results suggest that stratification by sex is important in the assessment of the recessive effect of the C-allele of rs13946 in ACL injury.

We also determined a specific haplotype of five SNPs for which the C-allele of rs13946 was exclusively found. This haplotype includes the C-allele of rs12722, also widely reported in association with ACL injury [28]. Accordingly, the lack of reproducibility of results in large cohorts may arise because studies may not be assessing the causal variant or a potential interaction among SNPs within the gene. Lulinska et al. [30] found that male football players homozygous for the C–C rs12722–rs13946 haplotype were protected against ACL injury. While our study suggests that the haplotype is a risk factor for females, further studies are needed in both males and females to confirm the sex-specific risk of the haplotype.

Overall, the present study suggests that genetic tests should be applied for sex-specific risk management and prevention to help elite female footballers to safely improve their performance. Physicians, trainers and therapists could benefit from risk-profiling athletes, including genetic, internal and environmental exposures, which can help in the design of individualized training regimes and strategies to prevent disabling injuries such as ACL rupture.

## 5. Conclusions

The study supports the role of SNPs in *COL5A1* as risk factors for non-contact ACL injury in female football players.

## Figures and Tables

**Figure 1 genes-14-00033-f001:**
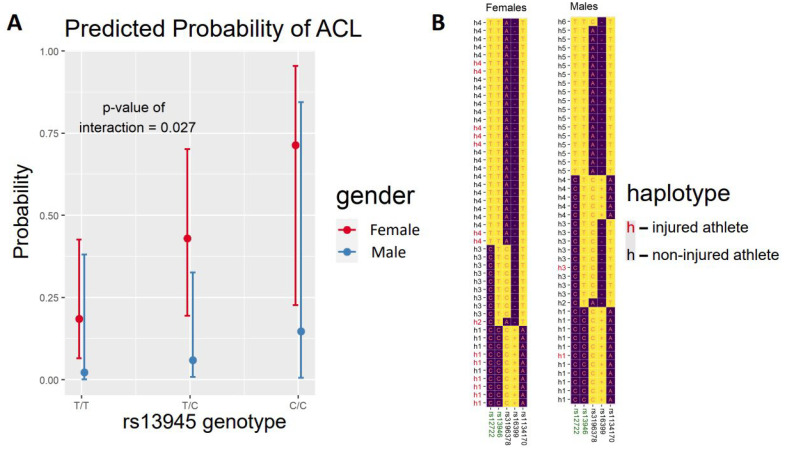
(**A**) Box-and-whisker plots of the probability of anterior cruciate ligament (ACL) rupture in female and male professional football players. (**B**) Haplotype inference on single-nucleotide polymorphisms (SNPs) spanning the *COL5A1* gene. Rows are individual haplotypes inferred from all the subjects in the study and are defined by the alleles (letters in the cells) of the SNPs in the gene (in columns).

**Table 1 genes-14-00033-t001:** Independent significant associations between SNPs at nominal level (*p*-value = 0.05) and anterior cruciate ligament (ACL) in elite female footballers. All significant associations were observed in collagen gene *COL5A1*.

	No	Yes	*p* Overall
	*N =* 17	*N =* 7	
rs13946:			0.017
T/T	13 (76.5%)	4 (57.1%)	
T/C	4 (23.5%)	0 (0.00%)	
C/C	0 (0.00%)	3 (42.9%)	
rs16399:			0.017
-/-	13 (76.5%)	4 (57.1%)	
CTAT/-	4 (23.5%)	0 (0.00%)	
CTAT/CTAT	0 (0.00%)	3 (42.9%)	
rs1134170:			0.017
T/T	13 (76.5%)	4 (57.1%)	
T/A	4 (23.5%)	0 (0.00%)	
A/A	0 (0.00%)	3 (42.9%)	
rs71746744:			0.017
GGGA/GGGA	13 (76.5%)	4 (57.1%)	
GGGA/-	4 (23.5%)	0 (0.00%)	
-/-	0 (0.00%)	3 (42.9%)	
rs3196378:			0.040
A/A	6 (35.3%)	4 (57.1%)	
C/A	9 (52.9%)	0 (0.00%)	
C/C	2 (11.8%)	3 (42.9%)	

## Data Availability

The data presented in this study are available upon request from the first author of this study.

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
