# Peer review of "Sex Differences in the Association between Risk of Anterior Cruciate Ligament Rupture and COL5A1 Polymorphisms in Elite Footballers"

_genes, 2022, doi:10.3390/genes14010033_

Round 1
Reviewer 1 Report
The appreciation of sex differences is essential for human health. The authors report a female-specific association between COL5A1 polymorphisms and the risk of non-contact anterior cruciate ligament (ACL) rupture with the study on a cohort of highly controlled male and female athletes.
Though there may be concerns about the limited sample size of participants, the authors successfully show evidence for the sex-specific ACL rupture risk-related SNPs. This study is of interest and well-conducted. Yet, there are two minor issues listed as follows:
1: There are inconsistent descriptions of the participants involved in the authors’ study: “Forty-six” in line 25, and “Forty-eight professional (24 women and 24 men) football players” in line 138.
2: Reference #17 might not directly support the statement in lines 78-80, please consider double-checking.
Author Response
Answer to reviewers’ comments.
Reviewer #1:
The appreciation of sex differences is essential for human health. The authors report a female-specific association between COL5A1 polymorphisms and the risk of non-contact anterior cruciate ligament (ACL) rupture with the study on a cohort of highly controlled male and female athletes.
Though there may be concerns about the limited sample size of participants, the authors successfully show evidence for the sex-specific ACL rupture risk-related SNPs. This study is of interest and well-conducted.
Answer: We thank the reviewer for the positive comment. We have addressed the minor issues in the new version of the manuscript.
Yet, there are two minor issues listed as follows:
1: There are inconsistent descriptions of the participants involved in the authors’ study: “Forty-six” in line 25, and “Forty-eight professional (24 women and 24 men) football players” in line 138.
Answer: We have corrected the figure in the new version. The correct number is 46.
2: Reference #17 might not directly support the statement in lines 78-80, please consider double-checking.
Answer: We thank the reviewer for spotting this mistake. We have corrected the reference.
Reviewer 2 Report
- The authors studies the association of several SNPs with the risk of anterior cruciate ligament (ACL) rupture in elite football players. I think the paper is well written, concise and address a relevant question given the potential of genetic profiling in prevention. I have some comments about this manuscript.
- Line 76: I guess you are referring to "twin studies" not "tween studies".
- Line 78: You say "A recent study of ACL injury in the general population observed three SNPs at a genome-wide significant level and replicated a previous association in a collagen gene COL3A1" but then cite a paper called "Cancer statistics". Are you sure this is the correct reference?
- Line 102: I think the sample is small, but as the authors explain in Discussion, the sample is very homogeneous and it is very interesting to have individuals at a high risk of ACL due to their activity. One thing not mentioned is the ancestry of the sample. It would be adequate to indicate if the sample is homogeneous in this regard or not. Although ancestry categorizations are arbitrary, they are still useful in this context given that mixed ancestries within a sample (e.g., European and African ancestries) could influence the results, so the reader should know that. If there is no information available, then this should be clearly stated in the text.
- Line 151: There is no mention to multiple test correction in the manuscript. I understand that the studied SNPs have been selected based on previous evidence, i.e., they have been already associated with ACL or related traits. In addition, the number of independent tests is not very high given the number of SNPs, the linkage disequilibrium and the combination of several inheritance models. But it would be useful for reader to have a mention about why the authors do not use methods like the Bonferroni correction for reducing the risk of false positives.
- Line 183: It could be useful for the reader to add some details or references about the methodology used to meta-analyze your results.
- Table 1: For rs16399, rs1134170 and rs71746744 are you showing haplotypes? This should be clarify in the legend.
Author Response
Answer to reviewers’ comments.
Reviewer #2:
-The authors studied the association of several SNPs with the risk of anterior cruciate ligament (ACL) rupture in elite football players. I think the paper is well written, concise and address a relevant question given the potential of genetic profiling in prevention. I have some comments about this manuscript.
Answer: We thank the reviewer for the positive comments. Please find below specific answers to the comments.
- Line 76: I guess you are referring to "twin studies" not "tween studies".
Answer: Corrected.
- Line 78: You say "A recent study of ACL injury in the general population observed three SNPs at a genome-wide significant level and replicated a previous association in a collagen gene COL3A1" but then cite a paper called "Cancer statistics". Are you sure this is the correct reference?
Answer: We thank the reviewer for spotting this mistake. We have corrected the reference.
- Line 102: I think the sample is small, but as the authors explain in Discussion, the sample is very homogeneous and it is very interesting to have individuals at a high risk of ACL due to their activity. One thing not mentioned is the ancestry of the sample. It would be adequate to indicate if the sample is homogeneous in this regard or not. Although ancestry categorizations are arbitrary, they are still useful in this context given that mixed ancestries within a sample (e.g., European and African ancestries) could influence the results, so the reader should know that. If there is no information available, then this should be clearly stated in the text.
Answer: We agree with the reviewer that this information is needed to help the reader contextualize the results. We have added some text at the end of section 2.2.
“Regarding ancestry, 23 female football players were of European origin and one of African ancestry. Whereas males were more diverse, 11 were from European ancestry, 3 from African origin and 8 from Latin America with likely admixture origin.”
- Line 151: There is no mention to multiple test correction in the manuscript. I understand that the studied SNPs have been selected based on previous evidence, i.e., they have been already associated with ACL or related traits. In addition, the number of independent tests is not very high given the number of SNPs, the linkage disequilibrium and the combination of several inheritance models. But it would be useful for reader to have a mention about why the authors do not use methods like the Bonferroni correction for reducing the risk of false positives.
Answer: Yes, the reviewer is right. The SNPs were selected from previous. Multiple comparison correction is a statistical test used to discover associations with no previous evidence and can hinder statistical power as stated in reference [26] we have added. We have made this clear in the new version of the manuscript and have added a new reference supporting this decision. At the end of the first paragraph of subsection 2.3, we now write:
“Since the SNPs were selected from previous evidence and the number of individuals of our study is low, we report nominal statistical significance, as multiple comparison correction is used to discover associations with no previous evidence and can hinder statistical power [26].”
Answer (cont.): Regarding correcting for the use of different inheritance models, we have done such correction by using the exact distribution of max-statistic (e.g. the most significant p-value of dominant, recessive and additive models) as described in Gonzalez et al. (https://pubmed.ncbi.nlm.nih.gov/18228557/). We have added the following paragraph to the methods section adding a new reference.
“We used the exact distribution of max-statistic (e.g the most significant p-value obtained from dominant, recessive and additive tests) to compute the p-value of association corrected for the number of inheritance models [27]”
Answer (cont.): The interaction analysis does not require multiple test corrections since we only analyze one SNP and used the additive model.
- Line 183: It could be useful for the reader to add some details or references about the methodology used to meta-analyze your results.
Answer: We agree with the reviewer that more description on the meta-analysis was required. We have now included a description at the end of subsection 2.3.
“We meta-analyzed the risk of ACL for men and women using our data and that reported in the meta-analysis of Guo et al. [26] from eight different studies. As reported, the studies had significant heterogeneity. Therefore, for adding our associations to the meta-analysis, we used Fisher’s method for the combination of P-values from the poolr R package (version 1.1-1).”
- Table 1: For rs16399, rs1134170 and rs71746744 are you showing haplotypes? This should be clarify in the legend.
Answer: To underline that these are not haplotypes but single SNP-wise associations, we have added in the table legend:
“Independent significant associations of SNPs”
Round 2
Reviewer 2 Report
The authors have done a good job answering my comments. Please accept.